# Hydrogels Based on Schiff Base Linkages for Biomedical Applications

**DOI:** 10.3390/molecules24163005

**Published:** 2019-08-19

**Authors:** Junpeng Xu, Yi Liu, Shan-hui Hsu

**Affiliations:** 1Institute of Polymer Science and Engineering, National Taiwan University, No. 1, Sec. 4 Roosevelt Road, Taipei 10617, Taiwan; 2Institute of Cellular and System Medicine, National Health Research Institutes, No. 35 Keyan Road, Miaoli 35053, Taiwan

**Keywords:** Schiff base, hydrogel, click chemistry, self-healing, dynamic covalent bond, tissue engineering

## Abstract

Schiff base, an important family of reaction in click chemistry, has received significant attention in the formation of self-healing hydrogels in recent years. Schiff base reversibly reacts even in mild conditions, which allows hydrogels with self-healing ability to recover their structures and functions after damages. Moreover, pH-sensitivity of the Schiff base offers the hydrogels response to biologically relevant stimuli. Different types of Schiff base can provide the hydrogels with tunable mechanical properties and chemical stabilities. In this review, we summarized the design and preparation of hydrogels based on various types of Schiff base linkages, as well as the biomedical applications of hydrogels in drug delivery, tissue regeneration, wound healing, tissue adhesives, bioprinting, and biosensors.

## 1. Introduction

Hydrogels with a three-dimensional structure and high water content are an important category of soft materials [1,2]. Hydrogels provide a physiologically similar environment for cell growth and they are often used to mimic the extracellular matrix (ECM) [3]. Cells interact and remodel with the ECM or ECM-like structures to achieve various cell behaviors, such as proliferation [4], migration [5], and differentiation [6]. Therefore, hydrogels have been developed for several biomedical applications [7], including tissue engineering [8], wound dressing [9], and drug delivery [10]. Hydrogels are generally formed by connecting hydrophilic polymer chains with various kinds of linkages to form networks [7]. The crosslinking network is divided into two types, i.e., the covalent network and non-covalent network, based on the nature of the linkages [11]. Click chemistry is one of the most common reactions used to obtain a covalent hydrogel network [12].

Click chemistry, initially proposed by Sharpless et al. in 2001, is a strategy combining various units to create highly selective products with a reliable high yield and effective atom utilization [13]. Click chemistry can yield smart materials that respond to external stimuli, and are often used as a key tool to prepare covalently crosslinked hydrogels for biomedical applications [14]. Typical examples of click chemistry are the Huisgen cycloaddition reaction [15], thiol-ene reaction [16], Diels–Alder reaction [17], Michael addition reaction [18], and the Schiff reaction [19]. These click reactions have been widely used in materials science [20,21,22], life sciences [23], and polymer chemistry [24,25]. Efficient click reactions in polymer chemistry have led to the development of new synthetic strategies. Click reactions can effectively create condensed polymers, as well as modify polymer chains and sophisticated architectures, including smart hydrogel networks or scaffolds. The Schiff reaction is a family of reactions used to generate dynamic Schiff base linkages, and it is currently one of the best methods to prepare smart biocompatible hydrogels.

The Schiff reaction is a chemical reaction involving a dynamic covalent imine bond formation via the crosslinking of amine groups and aldehyde groups [26]. Schiff base linkages can be generated in situ with cells, tissues, and bioactive molecules under physiological conditions to create hydrogel networks. The dynamic crosslinking networks can also give hydrogels their self-healing properties [27]. Moreover, the Schiff base is pH-responsive according to its chemical structure [28]. Schiff base linkages, including imines, hydrazones, and oximes, are the products of condensation reactions between aldehyde groups and various nucleophilic amine groups. Compared to imines, oximes and hydrazones have better chemical stability with pH value changes because the mesomeric effect reduces the electrophilicity of the original carbon–nitrogen double bond [28]. Hydrogels with various Schiff base linkages have unique properties to meet the requirements of different tissues. The Schiff reaction has excellent prospects in the biomedical field due to its simplicity, reversibility, pH-sensitivity, and biocompatibility.

Hydrogels with Schiff base linkages are currently used in several biomedical fields (Figure 1), including drug delivery [29,30], tissue regeneration [31,32], wound healing [33,34], tissue adhesives [35], bioprinting [36], and biosensors [37]. This review highlighted the use of different Schiff base linkages as crosslinking strategies to prepare biologically relevant hydrogels that can be used in physiological environments. After a brief classification of the Schiff base, we summarized the chemical mechanisms of different linkages. The preparation and biomedical applications of hydrogels based on the Schiff reaction were further epitomized.

## 2. Design and Preparation

The Schiff base formation is obtained via the nucleophilic attack of amines on the electrophilic carbon atoms of aldehydes or ketones. Hydrogels for biomedical applications have been extensively developed based on imines and their derivatives, including hydrazones and oximes. Imines, hydrazones, and oximes are formed by reaction between aldehydes/ketones and primary amines, hydrazides, and aminooxy, respectively. Hydrazones and oximes exhibit better intrinsic stability compared to imines. Furthermore, acylhydrazones allow higher hydrolytic stability compared to hydrazones and oximes [38]. In several recent studies, hydrazones and acylhydrazones with proper stability were employed to form crosslinks in hydrogel networks. Moreover, benzoic Schiff base linkages have been widely studied, where the stability of carbon–nitrogen double bonds has been improved through aromatic substitution with benzene rings connected to the carbon or nitrogen atoms [39]. Commonly, benzoic Schiff base linkages, including benzoic imines, benzoic hydrazides, and benzoic oximes, are formed via linkages of benzoic aldehydes with amines, hydrazides, and aminooxy’s, respectively.

Usually, chemical modifications are necessary to introduce aldehyde groups or nucleophilic groups to the polymers for hydrogel preparations based on imine, hydrazone, or oxime linkages. The polymers are functionalized with aldehyde groups through two main mechanisms: (1) The oxidative cleavage of vicinal diols with sodium periodate, and (2) the conjugation of aldehyde-containing molecules using carbodiimide chemistry. Periodate-mediated oxidation induces glycol cleavage to decompose into the aldehyde products shown in Figure 2. Periodate-mediated oxidation is generally used to cleave the vicinal diols of various polysaccharides, such as alginate [40,41,42,43], chondroitin sulfate [44], dextran [45,46,47,48], and hyaluronic acid [49,50,51]. Carbodiimide chemistry refers to amide or ester bond formation of carboxylic acids and amines or hydroxyls with the catalysis of carbodiimide compounds, such as water-soluble 1-ethyl-3-(3-dimethylaminopropyl) carbodiimide (EDC) and water-insoluble *N*,*N*′-dicyclohexylcarbodiimide (DCC). Aldehyde-containing molecules, such as 4-formylbenzoic acid, can be conjugated with the polymers through carbodiimide chemistry. The primary amine groups, as typical nucleophiles, exist in several natural and synthetic polymers. Hydrazide groups are conjugated with polymers through carbodiimide chemistry for hydrazone formation, whilst aminooxy groups are conjugated with polymers through the Mitsunobu reaction for oxime formation. The Mitsunobu reaction involves the dehydrative coupling of an alcohol to a pronucleophile, including carboxylic acids, phenols, imides, and sulfonamides, using triphenylphosphine and an azodicarboxylate.

### 2.1. Imine-Based Hydrogels

As shown in Figure 3A, imines refer to compounds with carbon–nitrogen double bonds. In several studies regarding imine-based hydrogels, polysaccharides (e.g., hyaluronic acid, chondroitin sulfate, alginate, and dextran) are modified with aldehyde groups through periodate-mediated oxidation, and then the oxidized polysaccharides are used to form imine-based hydrogels with various chitosan derivatives [41,42,44,45,51,52]. Liu et al. reported the preparation of hydrogels with tough and self-healing properties through conjugation of oxidized alginate and acrylamide (AM) via imine formation and subsequent radical polymerization between double bonds on conjugated AM moieties and free AM monomers [43]. The hydrogels showed better self-healing and mechanical properties compared to hydrogels prepared using a mixture of oxidized alginate and polyacrylamide. Recently, benzoic imines have attracted attention in biomaterials citing their self-healing ability. We introduce these researches in the following sections.

### 2.2. Hydrazone-Based Hydrogels

As shown in Figure 3B, hydrazones are formed by aldehydes/ketones and hydrazides. Figure 4A shows that hydrazide-functionalized polymers can be synthesized through conjugation of tri-Boc-hydrazinoacetic acid with amine groups on polymer chains and subsequent trifluoroacetic acid deprotection [49,53,54]. Wang et al. prepared an injectable hydrogel based on dynamic covalent hydrazone bonds between hydrazide-functionalized elastin-like protein and aldehyde-functionalized hyaluronic acid (HA) [53]. Tri-Boc-hydrazinoacetic acid was conjugated with elastin-like protein using a guanidinium (i.e., HATU, 1-[Bis(dimethylamino)methylene]-1*H*-1,2,3-triazolo[4,5-*b*]pyridinium 3-oxid hexafluorophosphate) coupling reagent. Injectability to the hydrogel was provided by dynamic hydrazone crosslinking. Meanwhile, elastin-like protein thermal aggregates reinforced the hydrogel with long-term degradation. McKinnon et al. reported an adaptable hydrogel that could respond to cell-induced stress by rapidly breaking and reforming reversible hydrazone bonds [55,56]. Hydrogels with a wide range of modulus and stress relaxation characteristics mimic the biophysics of native tissue and allow the functions of natural cells. Hydrazone-based hydrogels with dynamic tunability allow the development of cell morphologies, whereas non-adaptable hydrogels prevent cytoskeletal rearrangement and extension.

Acylhydrazones, analogous to hydrazones, are formed by aldehydes/ketones and acylhydrazides (Figure 3C). Acylhydrazones are widely applied in the preparation of hydrogels for biomedical applications via the reaction between aldehyde-containing polymers and dihydrazide-containing compounds or acylhydrazide-functionalized polymers [8,50,57,58,59]. Normally, acylhydrazide-functionalized polymers are synthesized using carbodiimide chemistry between carboxyl-containing polymers and dihydrazide-containing compounds, such as carbodihydrazide (CDH) and adipoyldihydrazide (ADH), similar to Figure 4B. Wang et al. presented a shear-thinning and self-healing hydrogel based on dynamic acylhydrazone bonds for three-dimensional (3D) printing [50]. In their study, acylhydrazine-functionalized HA was prepared via carbodiimide chemistry between carboxyl groups on HA and amine groups on ADH. The hydrogels, formed by mixing acylhydrazine-functionalized HA and aldehyde-functionalized HA, could be extruded and printed as 3D constructs with high fidelity and stability.

### 2.3. Oxime-Based Hydrogels

Oxime is obtained through a reaction between aldehydes/ketones and hydroxylamine, as shown in Figure 3D. Oxime-based hydrogels can be formed using aminooxy-functionalized polymers and aldehyde-functionalized polymers. Aminooxy-functionalized polymers can be synthesized in two steps using the Mitsunobu reaction, coupling *N*-hydroxyphthalimide with hydroxyl on the polymers, followed by deprotection with hydrazine (Figure 5) [8,47,60,61,62,63,64]. Grover et al. reported on oxime-based injectable hydrogels using aminooxy-functionalized four-arm poly(ethylene oxide) (AO-PEO) and ketone-functionalized four-arm poly(ethylene oxide) (Ket-PEO) [60]. For the synthesis of AO-PEO, triphenylphosphine and diisopropyl azodicarboxylate, respectively, were added into the mixture of alcohol-terminated four-arm PEO and *N*-hydroxyphthalimide in the cold condition. After a one day reaction, the product was mixed with hydrazine monohydrate. The hydrogels with pH-tunable gelation could be formed by AO-PEO and Ket-PEO, as well as AO-PEO and oxidized hyaluronic acid or oxidized alginate. Recently, photomediated oxime linkages were developed for spatiotemporally controlled hydrogel formation [65,66]. In these researches, 2-(2-nitrophenyl) propyloxycarbonyl (NPPOC) was utilized as a direct photocage for aminooxys. Upon UV irradiation, the aminooxys were freed from the NPPOC photocages and they reacted with aldehydes to form oxime linkages. Photomediated oxime linkages provided control over the location and extent of crosslinking for the systematic tuning of mechanical properties. Moreover, photomediated oxime linkages could be applied to immobilize various biomolecules with spatiotemporal control and micron-scale resolution.

### 2.4. Benzoic Schiff Base-Based Hydrogels

The benzoic Schiff base has attracted sizable attention in the preparation of hydrogels for biomedical applications. In addition to higher stability, the benzoic Schiff base gives hydrogels enhanced self-healing properties compared to an aliphatic Schiff base. Benzaldehyde-containing small molecules, such as zinc phthalocyanine tetra-aldehyde (ZnPcTa) [67,68] and 3,5-diformyl-4,4-difluoro-4-bora-3a,4a-diaza-s-indacene (3,5-diformyl-BODIPY) [69], have been developed as crosslinkers to generate benzoic Schiff base-based hydrogels. In most cases, benzaldehyde-functionalized polymers are synthesized to react with polymers containing amine groups. 

Benzaldehyde-terminated poly(ethylene oxide) (PEO) can be prepared through a nucleophilic substitution reaction followed by the synthesis of mesylate-terminated PEO, similar to Figure 6A [10,70,71]. First, mesylate-terminated PEO is synthesized through the reaction of PEO, trimethylamine, and methanesulfonyl chloride in methylene chloride solution. Then, 4-hydroxybenzaldehyde and potassium carbonate are added to the solution of mesylate-terminated PEO in dimethylformamide to obtain benzaldehyde-terminated PEO. Deng et al. reported a dynamic hydrogel with an environmental adaptive self-healing property via acylhydrazone linkages between benzaldehyde-terminated three-arm PEO and dithiodipropionic acid dihydrazide [59]. The hydrogel could automatically repair damages under acidic conditions through acylhydrazone exchange, whilst the hydrogel was not self-healable at pH 7 due to kinetically locked bonds. However, addition of catalytic aniline to the hydrogel facilitated hydrogel self-healing at pH 7 by accelerating acylhydrazone exchange. Yu et al. designed an injectable self-healing hydrogel based on a chain extended Pluronic F127 (PEO_90_-PPO_65_-PEO_90_) multi-block copolymer [71]. Benzaldehyde-terminated Pluronic F127 was synthesized and then extended using adipic dihydrazide via the formation of acylhydrazone bonds. The hydrogel exhibited rapid sol-gel transition at body temperature based on the thermo-responsivity of Pluronic F127. Moreover, dynamic acylhydrazone bonds allowed the hydrogel to recover its mechanical properties and structure after repeated damage.

Benzaldehyde-terminated PEO can also be facilely prepared in one step using carbodiimide condensation chemistry, similar to Figure 6B [37,72,73,74,75,76]. The very first work on the synthesis of benzaldehyde-terminated Pluronic L64 using carbodiimide chemistry was reported in 2010 [75]. The injectable hydrogels based on benzaldehyde-terminated Pluronic L64 and glycol chitosan were prepared under dual pH- and temperature-responsiveness. The benzaldehyde-terminated PEO has been widely used to form injectable self-healing hydrogels for various biomaterial applications. Benzaldehyde-terminated telechelic PEO was synthesized via the conjugation of 4-formylbenzoic acid with PEO chain ends catalyzed by DCC and 4-(dimethylamino)pyridine (DMAP) [73]. Benzaldehyde-terminated PEO was used to form a self-healing hydrogel with chitosan through dynamic Schiff base linkages. Moreover, the hydrogels showed multi-responsive behavior to pH, amino acids, and vitamin B6 derivatives because of the dynamic equilibrium between the Schiff base linkage and the aldehyde and amine reactants. Khan et al. prepared a benzaldehyde-terminated four-arm PEO using four-arm PEO and 4-formylbenzoic acid via an EDA coupling reaction [74]. An injectable self-healing hydrogel was generated using benzaldehyde-terminated four-arm PEO and chitosan-g-l-glutamic acid. The hydrogels exhibited tunable mechanical properties by varying the benzaldehyde-terminated four-arm PEO concentration or the total solid content of the hydrogels.

As shown in Figure 6C, a novel photogenerated aldehyde group (i.e., 2-nitrobenzyl alcohol group) has been developed to functionalize the polymers using carbodiimide chemistry [54,77,78]. 2-nitrosobenzaldehyde moieties can be converted by 2-nitrobenzyl alcohol moieties exposed to a 365 nm light wavelength. After generation of the benzaldehyde upon UV radiation, the phototriggered imine-crosslinked hydrogels would rapidly form. Yang et al. prepared a phototriggered imine-crosslinked hydrogel using hyaluronic acid-nitrobenzyl alcohol (HA-NB) and hyaluronic acid-carbohydrazide (HA-CDH) [77]. In this study, a 2-nitrobenzyl alcohol-containing molecule (i.e., *N*-(2-aminoethyl)-4-(4-(hydroxymethyl)-2-methoxy-5-nitrosophenoxy) butanamide) was synthesized to conjugate with HA for the preparation of photoreactive HA-NB. The hydrogel allowed excellent integration with surrounding tissue after in situ phototriggered crosslinking, owing to the imine formation between photogenerated aldehydes on HA-NB and amines on the tissue surface. 

## 3. Biomedical Applications

Hydrogels used in the biomedical field have several requirements, such as formation under physiological conditions, rapid crosslinking, and biocompatibility. In several studies, click chemistry, such as the Huisgen cycloaddition reaction, thiol-ene reaction, Diels–Alder reaction, and the Schiff reaction, has been developed to form crosslinks in hydrogel networks. However, metal catalysts, radical initiators, and poor reactivity tend to limit the Huisgen cycloaddition reaction, thiol-ene reaction, and Diels–Alder reaction, respectively [79]. Compared to other click reactions, the Schiff reaction can rapidly proceed under mild conditions without metal catalysts. Moreover, the Schiff reaction is used in biomaterials due to its reversibility, pH-responsiveness, high reactivity, and biocompatibility. A Schiff reaction with dynamic equilibrium naturally exists as one of the organisms self-healing mechanisms. In this section, we focused on the various applications of hydrogels with Schiff base linkages in the biomedical field.

### 3.1. Drug Delivery

One of the most important applications of hydrogels is drug delivery. Chemotherapy is clinically used against tumors and cancer, although challenges remain, such as inaccurate drug delivery and high initial dosage. Hydrogels are potential candidates to improve accuracy and efficiency, as well as reduce the adverse effects of chemotherapy. As drug carriers, hydrogels should have appropriate mechanical properties, injectability, biodegradability, and biocompatibility. Particularly, carriers must release the drug in a temporospatially appropriate manner. Hydrogels with Schiff base linkages are suitable to be designed and developed to satisfy application on various tissues citing their possibility to meet the aforementioned requirements.

Glycol chitosan-based hydrogels crosslinked with difunctionalized poly(*N*-isopropylacrylamide)-co-poly(acrylic acid) were prepared as drug delivery vehicles for anticancer chemotherapy [80]. Such hydrogels were designed to be injectable, self-healable, pH-sensitive, and temperature-sensitive. The release rate of cisplatin from such hydrogels was promoted in the acidic condition when the structure of the hydrogels collapsed. The controlled drug release behavior of the hydrogels in the corresponding pH environment suggested that these hydrogels could be used in drug delivery.

A self-healing hydrogel based on glycol chitosan and difunctionalized polyethylene glycol (DF-PEG) was loaded onto Taxol for antitumor therapy (Figure 7A) [81]. Compared to Pluronic F127 hydrogels, the injectable hydrogel released the drug for a significantly longer period. Tumor xenograft in nude mice also confirmed a better antitumor effect of the self-healing hydrogel compared to the drug only control group and the drug-loaded Pluronic F127 hydrogel (Figure 7B,C) (the animal experiments were performed in accordance with protocols approved by the ethics committee of Cancer Hospital, Chinese Academy of Medical Science). The results suggested that a self-healing hydrogel was good at the encapsulation and controlled release of drugs.

Another advantage of hydrogels with Schiff base linkages is that different chemicals can be conveniently loaded, considering that in vivo chemotherapies often involve the efficient synergistic effect of several drugs. For example, a dual-drug-loaded magnetic hydrogel (DDMH) with thermosensitivity and self-healing properties was prepared using glycol chitosan and DF-PEG for tumor therapy [82]. Doxorubicin (DOX), iron oxide, and docetaxel (DTX) were carried in the DDMH. DDMH could exhibit the potential for stimuli-responsive drug release and magnetic hyperthermia. DDMH proved to have good biocompatibility and asynchronous controlled release properties in cell experiments. At the same time, DDMH showed excellent thermal induction performance under an alternative magnetic field (AMF) to control the surrounding temperature (Figure 7D). The results of in vitro and in vivo antitumor experiments showed that the synergistic antitumor effect of hydrogels loaded with two chemicals was significantly enhanced compared to DOX or DTX/PLGA nanoparticle-loaded hydrogels, respectively (Figure 7E,F1,F2). In addition, the asynchronous controlled release of DOX and DTX and the controlled release by AMF-trigger in synchronous drug delivery systems were more effective in anticancer chemotherapy.

Embolization agents are essential for the treatment of various unresectable malignant tumors. A three-component dynamic self-healing hydrogel based on glycol-chitosan, carbazochrome, and DF-PEG was recently prepared to overcome the limitation of low therapeutic efficiency [30]. The controlled release of carbazochrome and the tunable decomposition of hydrogels were found to be achieved using DF-PEG with various concentrations. The three-component hydrogels could change the crosslinking density of the internal structure and control the release rate of carbazochrome by increasing the concentration of DF-PEG. The self-healing hydrogel could effectively treat the target embolization, and it had a low cost and risk for in vivo use. Additionally, the size and gelation time of the hydrogel could be precisely controlled to potentiate the hydrogel for targeting embolization therapy.

### 3.2. Wound Healing

Over the last century, hydrogels have been the focus of wound healing materials. Hydrogels can absorb and retain wound exudates, and at the same time, promote fibroblast proliferation and wound epithelium formation [83]. The internal structure of the hydrogel is very compact (the hole being 100 nm in full swelling state), which can effectively prevent bacteria from entering the wound and keep the wound moist [84,85]. Hydrogels have unique and adjustable mechanical properties to fit wounds of different tissues [86]. Compared to conventional bandages or gauzes, the high water retaining capacity of hydrogels makes them particularly soothing on wounds, and the non-sticky nature of hydrogels causes less discomfort in patients and less pain during peeling [87]. As a coolant for local wounds, it can also relieve pain and restore the degree of damage.

Recently, quaternized chitosan-g-polyaniline (QCSP) and benzaldehyde group functionalized poly(ethylene glycol)-co-poly(glycerol sebacate) (PEGS-FA) solution were mixed to prepare injectable self-healing hydrogel dressing [34]. Compared to traditional commercial dressings, the self-healing hydrogel dressing had better antibacterial activity, electric activity, and free radical scavenging ability (Figure 8A,B). Besides, it was possible to adjust the gelation process to regulate the rate of gelation, pore size, electrical conductivity, and swelling ratio of the self-healing hydrogel. Simultaneously, hemostatic properties could be obtained by adjusting the ratios of different QCSP and PEGS-FA. The efficiency of such hydrogels in repairing full-thickness skin wounds has been confirmed in animal experiments (Figure 8C).

Self-adaptive self-healing hydrogels based on chitosan and DF-PEG have been successfully developed [72]. Given the dynamic Schiff bases, the hydrogels can change their shape and move freely. Human internal tissues have difficulty in responding to external stimuli, while traditional intelligent biomaterials hardly change their shape and location to adapt to the internal oral environment. Hydrogels show better therapeutic effects in wound healing in vivo experiments relative to other control treatments, supporting the self-adapting effect of the materials on wound healing.

Hydrogels can also carry drugs and act in synergy with their properties on wound healing. A novel injectable hydrogel loaded with nano-curcumin based on *N*,*O*-carboxymethyl chitosan (CCS) and oxidized alginate (OA) was reported to accelerate skin wound repair [88]. Compared to commercial dressings and unloaded hydrogel controls, the nano-curcumin/CCS-OA hydrogel significantly decreased the wound area, and complete wound closure occurred after 14 days of treatment (Figure 8D,E). In vivo and in vitro experiments also confirmed that nano-curcumin could be released from the proposed hydrogel to enhance epidermal re-epithelialization and collagen deposition in wound tissue.

Injectable hydrogels of chitosan and oxidized konjac glucomannan designed using the Schiff reaction mechanism were successfully produced [9]. These injectable hydrogels had self-healing ability and biocompatibility, which could benefit wound healing. The physical properties (such as porosity and swelling ratio) could be changed by adjusting the crosslinking density. Meanwhile, excellent inhibition rates of *Staphylococcus aureus* and *Escherichia coli* could prevent wounds from external bacterial infections during the healing process. The injected hydrogel significantly shortened the wound healing time using a full-thickness skin defect model test. Additionally, histological examinations showed that the resultant hydrogel significantly accelerated the re-epithelialization of damaged tissues.

### 3.3. Tissue Regeneration

Hydrogels that can be injected into the body to promote tissue repair have become a popular topic, and they have also found several tissue engineering applications in recent years [89]. Hydrogels are considered to be biocompatible because of their structural similarities with macromolecules in vivo [7]. Hydrogels can also mimic many properties of the extracellular matrix in tissues, such as regulating cell functions and permitting the diffusion of nutrients, metabolites, and growth factors [90]. Using injectable hydrogels allows clinicians to transplant corresponding cells in a minimally invasive manner, and to a certain extent, promote tissue regeneration.

Self-healing hydrogels based on glycol chitosan and DF-PEG have been employed as a means to improve neural tissue regeneration [31]. Neural stem cell spheroids with an appropriate size (<300 μm) and embedded in the self-healing hydrogel were neurosphere-like. The neurosphere-like progenitors showed great proliferation and differentiation in the hydrogel, and they also had better efficiency in repairing the central neural system injury of zebrafish. 

An injectable chitosan-fibrin (CF) hydrogel with an interpenetrating polymer network can be prepared by adding fibrin into a glycol chitosan-based hydrogel [32]. Vascular endothelial cells in CF hydrogels may form capillary-like structures. The injection of CF hydrogel alone promoted angiogenesis in the perivitelline space of zebrafish (Figure 9A). The CF hydrogel also proved to successfully repair the defects in ischemic hindlimb mice, as well as rescuing blood circulation (Figure 9B,C) (the animal experiments were performed in accordance with protocols approved by the University Animal Care and Use Committee).

Adding conductive polymers to hydrogels is a common way to generate conductive hydrogels, which can benefit electroactive cells and tissues. The combination of dextran-graft-aniline tetramer-graft-4-formylbenzoic acid (Dex-AT) and *N*-carboxyethyl chitosan (CECS) hydrogels was performed as a conductive injectable self-healing hydrogel [91]. Given the dynamic action of the Schiff base, these conductive hydrogels had good injectability and degradability. L929 fibroblasts could proliferate in the hydrogel. C2C12 myoblasts and human umbilical vein endothelial cells that were grown in the hydrogel could be released from the hydrogel matrix with a linear profile, and the released cells still showed an ability for continuous proliferation. Compared to the PBS control group and Dex/CECS hydrogel, the regeneration of skeletal muscle using a Dex-AT/CECS hydrogel was confirmed in an animal model within four weeks, as shown in Figure 9D,E.

Cell therapy is a potential strategy for tissue regeneration in heart disease. To effectively carry cells, biocompatible hydrogels can be a proper candidate for cell therapy. A conductive hydrogel based on chitosan-graft-aniline tetramer (CS-AT) and DF-PEG was prepared [92]. The conductivity of the prepared hydrogel was close to that of natural heart tissue, and it showed good tissue adhesion and antimicrobial activity to the target tissue. C2C12 myoblasts and H9c2 cells encapsulated in such hydrogels determined good viability and proliferation. Meanwhile, the injectability and biodegradability of hydrogels were proven through subcutaneous injection and in vivo degradation. Collectively, the experimental results showed that the hydrogel was an excellent cell carrier for cardiac tissue regeneration.

*N*-succinyl-chitosan (S-CS) and aldehyde-hyaluronic acid (A-HA) based composite hydrogels were successfully prepared, and they showed good biocompatibility [93]. As the ratio of S-CS to A-HA increased, the compressive modulus of the composite hydrogel improved, and the in vitro degradation rate slightly increased. The S-CS/A-HA hydrogels may promote cell adhesion and proliferation, as well as maintain regular cell morphology of bovine articular chondrocytes, demonstrating potential applications in cartilage tissue engineering. 

### 3.4. Bioprinting

Regarding the manufacturing process for tissue engineering applications, combining three-dimensional (3D) bioprinting technology with bioink, a substance containing living cells and materials to support the adhesion, proliferation, and differentiation of cells can generate 3D scaffolds with a tissue-like structure. The tissue-like 3D scaffold can provide structural support for cells at both a micro- and macro-level to improve the characteristics of the growth environment [94]. Meanwhile, such 3D scaffolds can be designed using computer programming and controlling printing parameters to achieve the requirements of high resolution, multi-length scale, and customization [95]. 3D bioprinting has been extensively studied in the biomedical field citing in situ cell mixing, rapid manufacturing, and the potential for artificial vessels and organs.

In recent years, hyaluronic acid-based hydrogels have been successfully prepared to contain dynamic hydrazone bonds using modified hyaluronic acid with hydrazide or aldehyde groups [50]. These hydrogels can be used to print 3D structures with high shape fidelity, relaxation stability, and cell compatibility due to the shear thinning and self-repairing properties of hydrogels with dynamic bonds (Figure 10A). The extruded hydrogel filaments printed with cells in situ have a smooth surface, high quality of reproduction, and no relaxation in the printing process (Figure 10B). Further complex structures can be successfully printed with high cell viability (Figure 10C).

Aldehyde-containing oxidized alginate was recently developed as a base material to develop self-repairing and 3D bioprintable hydrogels with different kinds of crosslinking agents (Figure 10D) [36]. Different imine dynamic covalent bonds (oxime, semicarbazide, and hydrazone) were used to control the hardness, viscoelasticity, self-healing properties, 3D printability, and morphological changes of cells in the hydrogels. Compared to oxime-bonded hydrogels, semicarbazide and hydrazone-bonded hydrogels are viscoelastic, self-healable, and 3D bioprintable. Subsequently, human dermal fibroblasts in printed hydrogels kept growing over seven days. The viscoelastic crosslinking (urea and hydrazone) was beneficial to cell extension and growth, whilst elastic crosslinking (oxime) restricted the morphology of human dermal fibroblasts to remain in a circular shape.

Water-soluble hydroxybutyl chitosan (HBC) and chondroitin oxysulfate (OCS) were used to prepare bioink based on the Schiff-base reaction to print different structures of hydrogels using different sacrificial molds with 3D bioprinting technology [96]. After the formula optimization, the injectable hydrogels with uniform internal holes (100 μm) were printed to form macroporous hydrogels. The biocompatible hydrogel constructs with various internal and external structures promoted the viability of cultured human adipose-derived mesenchymal stem cells. The bionic HBC/OCS hydrogels with controllable shape could be optimized and customized for specific cartilage engineering applications. Therefore, bioprintable HBC/OCS hydrogel has applications in tissue engineering.

### 3.5. Tissue Adhesives

With the rapid growth of the wound care market, tissue adhesives have been well studied and developed. In surgery, tissue adhesives can be used to replace conventional surgical adhesives due to their sufficient tensile strength, short bonding time, simple usage, and no need for postoperative removal [97]. Furthermore, the use of tissue adhesives in organisms needs to consider tissue toxicity given direct contact with body fluids or blood. Meanwhile, adhesives need to show the corresponding physical and chemical properties for different regions and tissues, fast adherence to target sites, and rapid hemostasis [35,98]. Therefore, biomedical adhesives have more stringent conditions related to toxicity and harmfulness, as well as strict biocompatibility and biodegradability standards. Hydrogels, as materials with excellent potential for tissue engineering, can be designed to achieve the aforementioned conditions for biomedical tissue adhesives across various applications by employing different materials and formulations.

Previously, a double crosslinked network (DN) hydrogel based on the combination of HA and furan derivatives using a Diels–Alder (DA) click reaction and acylhydrazone bond was successfully prepared [57]. Dynamic covalent acylhydrazone bonds can impart hydrogels with self-healing properties and control the crosslinking density of the network. Aldehyde groups in hydrogels based on the Schiff reaction can promote the adhesion of hydrogels to surrounding tissues. In contrast to non-sticky DA hydrogels, push-out tests were used to determine the bonding strength between DN hydrogels and cartilage. The results showed that the bond strength of the target hydrogel (10.3 ± 0.7 kPa) was significantly higher than that of the DA hydrogel (1.2 ± 0.5 kPa), which supported the tissue adhesive ability of the DN hydrogel.

In recent years, injectable double crosslinked self-healing hydrogels based on dopamine-grafted oxidized sodium alginate (OSA-DA) and polyacrylamide (PAM) have been reported for wound healing (Figure 11A) [99]. Given the hydrogen bond and Schiff base bond, the OSA-DA-PAM self-healing hydrogel has stable mechanical properties, such as high tensile strength (0.109 MPa) and elongation (2550%). Additionally, a large number of catechol groups on the OSA-DA chain may endow the hydrogel with unique cell affinity and tissue adhesion. The above properties of OSA-DA-PAM hydrogels are demonstrated in Figure 11B,C using animal experiments with robust wound protection and the ability to promote tissue regeneration.

A self-healing injectable micellar hydrogel can be successfully prepared (Figure 11D) under physiological conditions using quaternized chitosan (QCS) and benzaldehyde-terminated Pluronic F127 (PF) [70]. QCS/PF hydrogels exhibit appropriate modulus that is similar to human soft tissue, as well as stable rheological properties, excellent cell adhesion, and biocompatibility. Simultaneously, the resultant hydrogel also has pH response characteristics showing a high drug release rate in an acidic skin environment. The inherent antimicrobial property, free radical scavenging ability, and good coagulation ability of the hydrogel can effectively promote wound healing. Moreover, animal experiments validated the faster healing ability of the QCS/PF hydrogel compared to commercial wound adhesive dressings (Tegaderm^TM^). Besides, QCS/PF hydrogel loaded with curcumin (Cur-QCS/PF hydrogel) significantly promotes wound healing by up-regulating the production of wound healing-related factors and down-regulating the production of pro-inflammatory factors. Cur-QCS/PF hydrogel can also be applied to the human elbow citing its tissue adhesive ability (Figure 11E).

Hydrogels based on aminated star-shaped polyethylene glycol and aldehyde-dextran polymers have been reported as tissue adhesives [100]. Hydrogels with various cohesiveness, adhesiveness, and other physical properties can be designed by controlling the aldehyde/amine ratio, which can produce materials that are adhesive to specific tissues. The adhesive hydrogel has also shown good biocompatibility and tissue stickiness during in vivo and in vitro studies.

Two-component hydrogels based on aldehyde-dextran and dendritic polyamidoamine dendrimer were successfully prepared as tissue adhesive materials [101]. The viscous hydrogel provides viscosity via the crosslinking between aldehyde and amine groups, and it produces an adhesive interface in vivo by reacting with histamine glucan aldehyde. During material design, the interface morphology, bonding strength, and bonding mechanical properties of the hydrogel can be adjusted by controlling the aldehyde/amine ratio to achieve the specific microenvironment conditions. Good biocompatibility, tunability, and tissue adhesion were supported in cell experiments and mouse duodenal models.

### 3.6. Biosensors

Hydrogels, as a self-adhesive and elastic soft material with great potential in biomedical applications, can be used for wearable applications, including electronic skin and biosensors after being endowed with electrical conductivity [102]. Conductive hydrogels are typically based on conductive polymers with intrinsic electron conductivity, as well as conductive nanomaterials, such as polypyrrole (PPy), polythiophene (PT), polyaniline (PANI), carbon nanotubes, and metallic nanoparticles [103]. These different conductive hydrogels act on different fields depending on their strength, electrical conductivity, and stimuli responsiveness. In recent years, biosensors have been vigorously developed. 

A dynamic covalent hydrogel, based on an imine bond prepared using human neutrophil elastase (HNE), Ala-Ala-Pro-Val-Ala-Ala-Lys, and aldehyde-dextran was successfully reported [104]. In this study, hydrogel membranes were prepared by coating, which led to rapid selective degradation when enzymes were added. In the presence of periodontal disease markers, the activity of HNE and cathepsin could be directly measured by monitoring the degradation of the membranes. The low enzyme concentration could also be detected by increasing the crosslinking density of the peptide to increase the degradation rate (i.e., sensitivity).

A conductive hydrogel based on graphene oxide, dopamine, and polyacrylamide was prepared using the Schiff reaction, as shown in Figure 12 [37]. The high stretchability, toughness, and self-adherence of the conductive hydrogel provided great benefits as a biosensor. Wrist pulses were successfully detected through instantaneous changes in the electrical current and resistance using the skin attachment test. Hydrogels had good self-absorption ability which was similar to natural tissues, and good cell compatibility, allowing them to be implanted in vivo without causing any inflammation. The conductive hydrogel sensor was implanted into rabbits, and the signal for muscle fluctuation was easily obtained, supporting the potential for clinical applications (the animal experiments were performed in accordance with protocols approved by the local ethical committee and laboratory animal administration rules of China).

A hydrogel membrane assembled layer by layer from partially aldehyde-dextran, chitosan, and glucose oxidase (GOD) was reported for sensing glucose changes [105]. Since the GOD in the membrane converted the external glucose into gluconic acid, which reduced the local pH value of the membrane and triggered the expansion of the pH-sensitive membrane. Fabry–Perot fringes were induced by the swelling of the hydrogel membrane caused by glucose, which showed the change of glucose concentration. GOD operated well under physiological pH conditions, exhibiting high anti-interference, and no response to potential interferences. Particularly, the hydrogel sensing membrane showed a linear response within the clinically relevant glucose range, and the response speed was also fast. The resultant glucose biosensor may have the potential for real-time and continuous glucose monitoring.

## 4. Summary and Outlook

Hydrogels constructed using dynamic Schiff base linkages have been widely used in the biomedical field because of their non-stimulated uncoupling and recoupling under mild conditions. In this review, we focused on the categories, chemical mechanisms, and differences between the various types of Schiff base linkages, and we introduced the applications of hydrogels formed by the Schiff base in the biomedical area, including drug delivery, wound healing, tissue regeneration, bioprinting, biosensors, and tissue adhesives. 

Various types of Schiff base linkages have applied in hydrogel formations. Imine, hydrazone, acylhydrazone, or oxime are formed through reactions between primary amine, hydrazine, acylhydrazine, or aminooxy and aldehyde. Amongst imines and their derivatives, acylhydrazones exhibit the best hydrolytic stability. Moreover, the bond stability of imines can be improved by connecting the benzene rings to the carbon or nitrogen atoms of the carbon–nitrogen double bonds. Therefore, Schiff base formations between benzoic aldehydes and acylhydrazines have gained significant attention in recent researches. Primary amine groups exist across a number of natural polymers, while chemical functionalizations are usually necessary to introduce the aldehyde groups or nucleophilic groups to the polymers. Periodate-mediated oxidation of polysaccharides is the most common method to introduce aldehyde groups through the cleavage of vicinal diols. Hydrazide and aminooxy groups can be conjugated with polymers through chemical modifications for hydrazone and oxime formations, respectively.

Such hydrogels based on the Schiff reaction have self-healing ability, injectability, and other smart properties, owing to the reversibility and rapidity of the Schiff reaction. The Schiff reaction naturally exists in the human body and it plays an important role in the self-healing ability of human tissues. Citing strain sensitivity, biocompatibility, and self-healing properties, hydrogels composed of Schiff base linkages can be injected in situ and can integrate as a whole to display their various functions in minimally invasive and precise medical treatments, without the limitation of location or shape. Meanwhile, the aldehyde groups in the hydrogel can also bind to some integrins containing amine groups in human tissues, which can better combine with the host tissues. All the above properties show that smart hydrogels have great potential for applications in the biomedical field.

Understanding the chemical mechanisms of different Schiff base linkages and the crosslinking process of hydrogels is of great significance to the design of smart hydrogels for biomedical applications. Hydrogels based on the Schiff reaction with universal self-healing properties have good biocompatibility, injectability, and easy functionalization, which has effective applications in drug delivery, bioprinting, tissue adhesives, and wound healing. Meanwhile, several research teams are gradually introducing innovative elements into the dynamic hydrogel network, and they are developing multi-component smart hydrogels with self-healing properties, conductivity, magnetism, optical, or thermal effects. Such smart hydrogel systems will have wide applications and prospective uses in cancer treatment, tissue regeneration, medical imaging, soft actuators, or artificial organs.

## Figures and Tables

**Figure 1 molecules-24-03005-f001:**
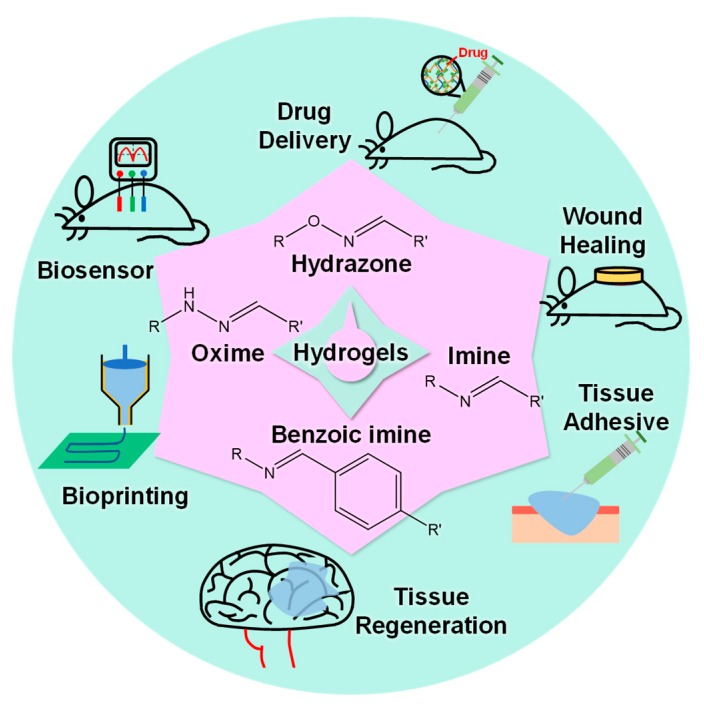
Summary of biomedical applications for hydrogels based on various Schiff base linkages.

**Figure 2 molecules-24-03005-f002:**
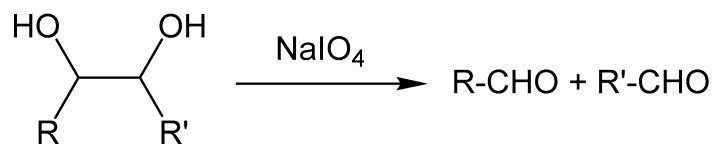
Periodate-mediated oxidation inducing the cleavage of vicinal diols.

**Figure 3 molecules-24-03005-f003:**
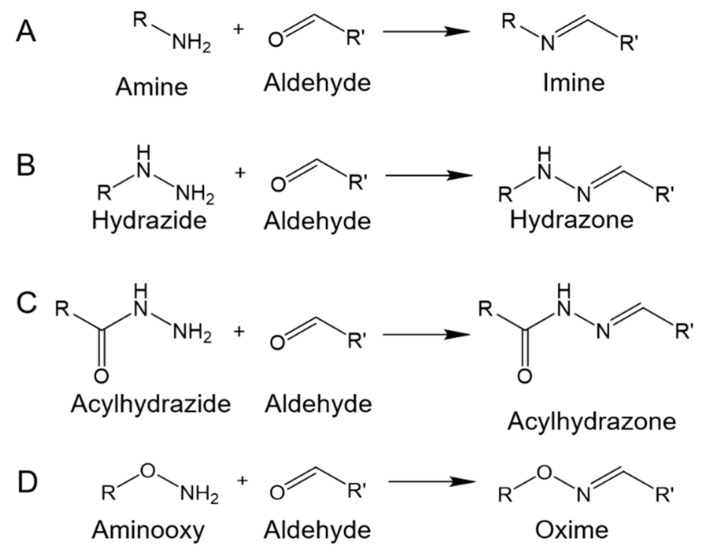
Formation of imine, hydrazone, acylhydrazone, or oxime through reactions between primary amine, hydrazide, acylhydrazide, or aminooxy and aldehyde.

**Figure 4 molecules-24-03005-f004:**
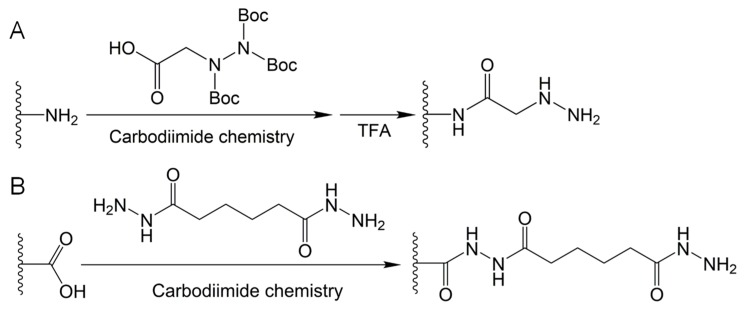
(**A**) Synthesis of hydrazide-functionalized polymer using carbodiimide chemistry coupling with tri-Boc-hydrazinoacetic acid for amide formation, and subsequent trifluoroacetic acid (TFA) deprotection of *tert*-butoxycarbonyl (Boc) groups. (**B**) Synthesis of acylhydrazide-functionalized polymer using carbodiimide chemistry coupling with dihydrazide-containing compounds, for example, adipoyldihydrazide coupling with a carboxyl group by amide formation.

**Figure 5 molecules-24-03005-f005:**
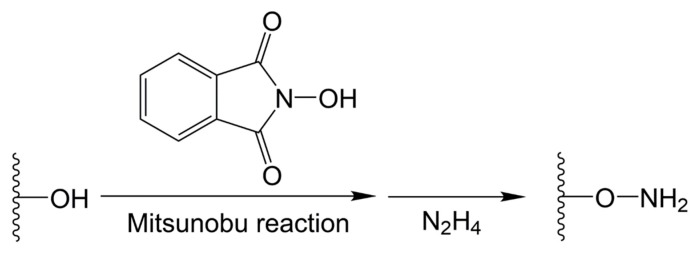
Synthesis of aminooxy-functionalized polymers using the Mitsunobu reaction coupling with hydroxyphthalimide, followed by deprotection with hydrazine (N_2_H_4_).

**Figure 6 molecules-24-03005-f006:**
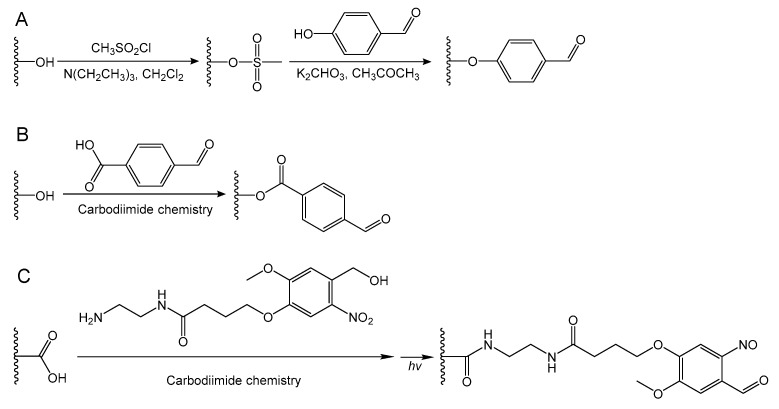
Synthesis of benzaldehyde-functionalized polymers through (**A**) a nucleophilic substitution reaction after the conversion of an alcohol to a mesylate; or carbodiimide chemistry coupling with the molecular containing benzaldehyde groups (**B**) or 2-nitrobenzyl alcohol groups (**C**). Photoreactive 2-nitrobenzyl alcohol can convert to 2-nitrosobenzaldehyde upon UV irradiation.

**Figure 7 molecules-24-03005-f007:**
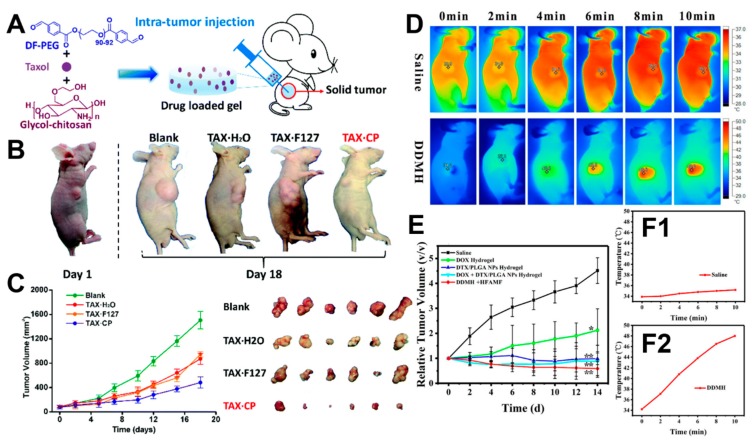
(**A**) Schematics of the intratumor experiment through hydrogel injection. (**B**) Results of the intratumor injection in nude mice divided into four groups according to different injection treatments, including only injected saline (Blank), only injected Taxol solution (Tax·H_2_O), injected Pluronic F127 hydrogel with Taxol (Tax·F127), and injected self-healing hydrogel based on glycol chitosan and DF-PEG with Taxol (Tax·CP) (the animal experiments were performed in accordance with protocols approved by the ethics committee of Cancer Hospital, Chinese Academy of Medical Science). (**C**) The volume and appearance of tumors in mice in the four groups. Reprinted with permission, as in Reference [81]. © 2017, Royal Society of Chemistry (London, UK). (**D**) Infrared thermal images for tumor-bearing mice under exposure to an AMF at different post-injection treatments (saline and DDMH) (the animal experiments were performed in accordance with protocols approved by the Laboratory Animal Research Center, Tsinghua University). (**E**) Tumor growth curves with different treatments (only saline, only DOX solution, DTX solution with PLGA nanoparticles, DOX solution with PLGA nanoparticles, and DDMH solution with AMF). (**F1**,**F2**) The temperature of different tumor sites shown during the exposure. Reprinted with permission [82]. © 2015, American Chemical Society (Washington, DC, USA).

**Figure 8 molecules-24-03005-f008:**
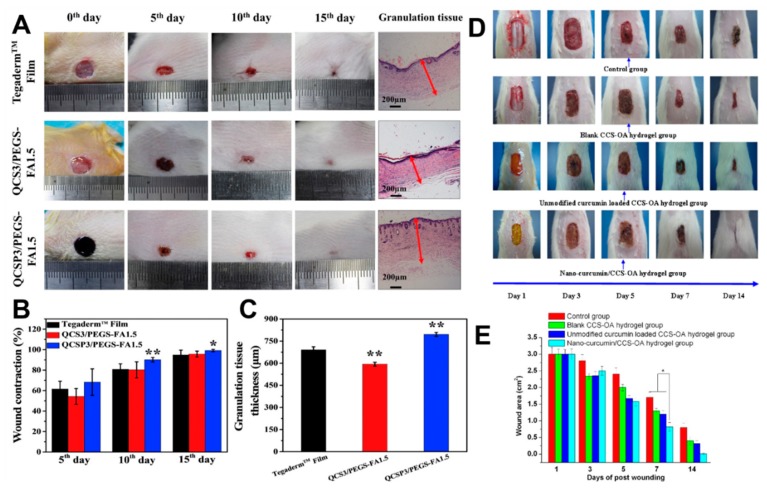
(**A**) Photographs of wounds at the 0th, 5th, 10th, and 15th day, and granulation tissue on the 15th day for commercial film dressings (Tegaderm™), hydrogel QCS/PEGS-FA, and hydrogel QCSP/PEGS-FA. (**B**) Wound contraction for the three groups. (**C**) Granulation tissue thickness for the three groups on the 15th day. Reprinted with permission [34]. © 2017, Elsevier Ltd. (Amsterdam, The Netherlands). (**D**) Photographs of wound treated over 14 days with saline solution, blank CCS-OA hydrogel, unmodified curcumin loaded CCS-OA hydrogel, and nano-curcumin/CCS-OA hydrogel. (**E**) The average wound area on three rats in each group (the animal experiments were performed in accordance with protocols approved by the Institutional Animal Care and Use Committee of Wenzhou medical college). Reprinted with permission [88]. © 2012, Elsevier Ltd. (Amsterdam, The Netherlands).

**Figure 9 molecules-24-03005-f009:**
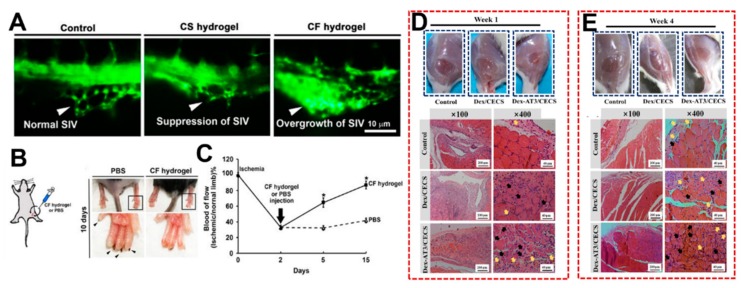
(**A**) The pattern of the sub intestinal vessels (SIV) in zebrafish embryos in which all of the vascular endothelial cells expressed enhanced green fluorescent protein. Each embryo was injected with PBS, CS hydrogel or CF hydrogel without cells 24 h after fertilization, and the fluorescence microscopic images were taken one day after the injection. (**B**) Representative images of the ischemic hind limbs of mice injected with PBS or CF hydrogel (the animal experiments were performed in accordance with protocols approved by the University Animal Care and Use Committee). The PBS group suffered severe nail loss. No intramedullary nail fell off in the CF hydrogel group and limb salvage was successful. (**C**) The blood flow ratios of ischemic limbs and healthy limbs in different groups showed a significant increase in blood flow after surgery in the hydrogel injection group. Reprinted with permission [32]. © 2017, Nature Publishing Group (London, UK). The regeneration of skeletal muscle tissue was evaluated using H&E staining in a rat skeletal muscle volume loss model after volumetric muscle loss injury and treatment for 1 week (**D**) and 4 weeks (**E**) between three groups, including PBS control, Dex/CECS hydrogel treatment, and Dex-AT/CECS hydrogel treatment (black arrows: centronucleated myofibers; yellow arrows: newly formed blood vessels). Reprinted with permission [91]. © 2019, Elsevier Ltd. (Amsterdam, The Netherlands).

**Figure 10 molecules-24-03005-f010:**
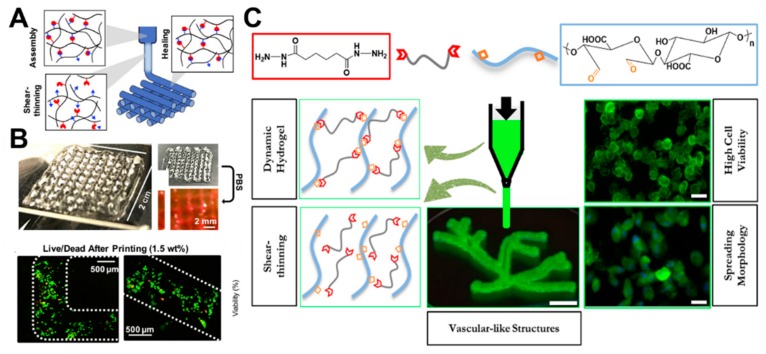
(**A**) Schematics for shear-thinning and self-healing of hydrogels during printing, where the hydrogels were formed in syringes, and filaments were stabilized by the self-healing properties of hydrazone bonds through shearing during the extrusion process. (**B**) Photos of 4-layer lattices in air and in PBS. (**C**) Live/dead staining of cells in lattice filaments immediately after extrusion. Reprinted with permission [50]. © 2018, Wiley-VCH (Weinheim, Germany). (**D**) The schematic diagram for the chemical structure and advantages of the bioprintable hydrogels [36].

**Figure 11 molecules-24-03005-f011:**
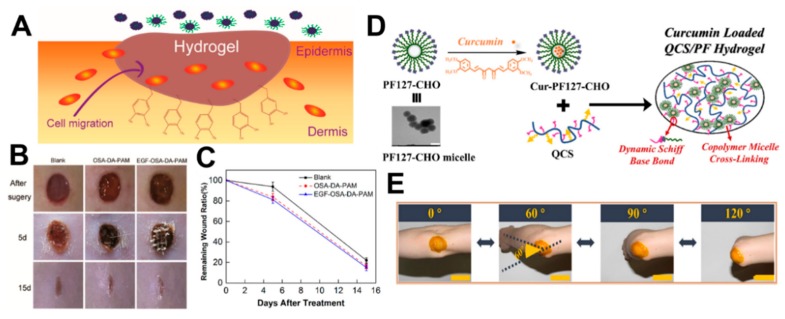
(**A**) The schematic diagram of tissue adhesive hydrogels. (**B**) Optical images of wound sites were taken on days 0, 5, and 15 after surgery. (**C**) Remaining wound ratios for each group. Reprinted with permission [99]. © 2018, American Chemical Society (Washington, DC, USA). (**D**) The schematic illustration of the Cur-QCS/PF hydrogel and the transmission electron microscope image of PF (i.e., PF127-CHO) micelles. Scale bar: 200 nm. (**E**) The photograph of Cur-QCS/PF hydrogels that were applied on the human elbow. Scale bar: 5 cm. Reprinted with permission [70]. © 2018, Elsevier Ltd. (Amsterdam, The Netherlands).

**Figure 12 molecules-24-03005-f012:**
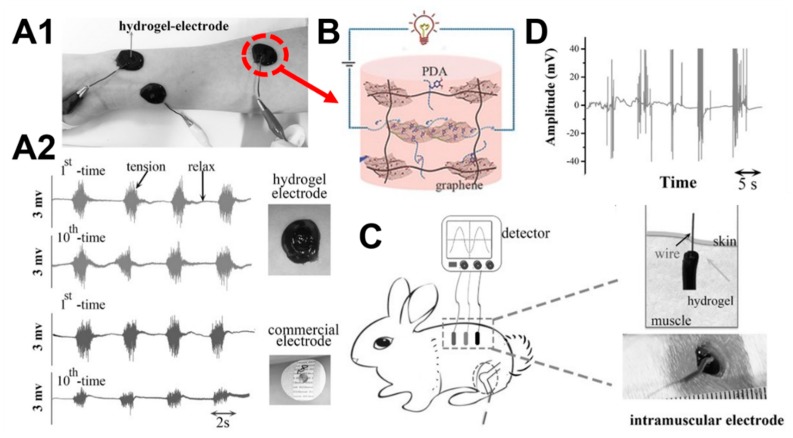
Conductive hydrogel as self-adhesive electrode biosensors. (**A1**) The hydrogel self-adhered to human arm skin and detected the electromyographic signals. (**A2**) Comparisons of electromyography recorded by hydrogel electrodes and commercial electrodes after ten cycles of usage (adhere-strip). (**B**) The schematic diagram of the chemical structure of conductive hydrogels. (**C**) Three hydrogel electrodes were implanted onto the dorsal muscle, and the wires from the electrodes were transcutaneously connected to the signal detector. (**D**) Example of electromyography recorded by the implanted hydrogel electrode biosensors from the muscle when the target rabbit was exposed to external stimuli (the animal experiments were performed in accordance with protocols approved by the local ethical committee and laboratory animal administration rules of China). Reprinted with permission [37]. © 2016, Wiley-VCH (Weinheim, Germany).

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
