# Peer review of "Hydrogels Based on Schiff Base Linkages for Biomedical Applications"

_molecules, 2019, doi:10.3390/molecules24163005_

Round 1
Reviewer 1 Report
The review gives a nice overview of hydrogels prepared by Schiff base reaction and their main applications. The topic is of interest but some parts of the text sounds repetitive and not very clear.
Please find below some points for revision:
The abstract and introduction need some rewriting, especially lines 38-42, 45-47, 51-54.
Figure 4 is confusing: I would suggest to a) remove the text ‘carbodiimide chemistry’ and explain that the amino or carboxylic groups containing molecules can be conjugated to a polymer with complementary functional groups via carbodiimide chemistry; or b) just write the name of the molecules and explain the reaction in the figure caption.
In text is mentioned several times carbodiimide chemistry but is never explained how it works or why it is used.
Line 207: it is not clear what the authors mean by stability.
Line 227-235 needs some rewriting.
Bioprinting paragraph: please correct the definition of bioink and rewrite the all paragraph. Also, in one of the last examples it is mentioned that the cells are round. It should be clarified which type of cells were used, for example for fibroblast is not a good indication if they maintain a round shape.
Author Response
It is very much appreciated that the reviewers made valuable comments. Our replies to the comments and the improvement of the manuscript based on the comments are listed below.
---------------------------------------------------------------------------------------
Reviewers’ Comments to Author:
1. The abstract and introduction need some rewriting, especially lines 38-42, 45-47, 51-54.
Ans: Thank you for the suggestion. We have already rewritten these parts and further proofread to avoid errors/poor sentences. We have highlighted the modified parts in lines 40-44, 48-58.
2. Figure 4 is confusing: I would suggest to a) remove the text ‘carbodiimide chemistry’ and explain that the amino or carboxylic groups containing molecules can be conjugated to a polymer with complementary functional groups via carbodiimide chemistry; or b) just write the name of the molecules and explain the reaction in the figure caption.
Ans. Thanks for your suggestions. We have written the name of the molecules and explained the reaction in the figure caption according to your suggestion (b).
3. In text is mentioned several times carbodiimide chemistry but is never explained how it works or why it is used.
Ans. Carbodiimide chemistry is widely used as a conjugation technique for modifying or crosslinking polymers. “Carbodiimide chemistry” is a conventional terminology meaning amide or ester bond formation by carboxylic acids and amines or hydroxyls with the catalysis of carbodiimide compound. Carbodiimide chemistry provides the most popular and versatile crosslinking method, and is considered a zero-length carboxyl-to-amine crosslinker. The mechanism of carbodiimide chemistry is shown in the figure below (https://www.thermofisher.com/tw/zt/home/life-science/protein-biology/protein-biology-learning-center/protein-biology-resource-library/pierce-protein-methods/carbodiimide-crosslinker-chemistry.html#1). Wherein, EDC is one of carbodiimide compounds, and sulfo-NHS is added to the reaction to increase efficiency and enable molecule ① to be activated for storage and later use.
4. Line 207: it is not clear what the authors mean by stability.
Ans: We have already modified this part according to your advice in line 247-248.
5. Line 227-235 needs some rewriting.
Ans: Thank you for the suggestion. We have already rewritten these parts in lines 265-273 and further proofread to avoid errors/poor sentences.
6.Bioprinting paragraph: please correct the definition of bioink and rewrite the all paragraph. Also, in one of the last examples it is mentioned that the cells are round. It should be clarified which type of cells were used, for example for fibroblast is not a good indication if they maintain a round shape.
Ans: Thank you for the suggestion. We have already rewritten the first paragraph of the bioprinting section and correct the definition of “bioink” in lines 409-412. Human dermal fibroblasts are grown in the oxime-elastic crosslinking network did not show the normal morphology in the penultimate example of bioprinting section. We have already rewritten this sentence in lines 431-434 following your suggestion.
Reviewer 2 Report
The review paper summarizes the works on hydrogels containing Schiff base linkages. The current version of the manuscript cannot be accepted for publication because several similar reviews, i.e. reporting on imine linker in polymer chemistry and biomedical applications, have been published, besides the quality of the manuscript itself is far from satisfied.
1. The authors classified Schiff’s reaction under the catalog of Click chemistry. This might be wrong. Click chemistry represents the selective reactions that can rapidly join small modular units together in high yield and without offensive byproducts. In most of the cases, Schiff’s reaction is reversible, that is, the formation and hydrolysis are in dynamic balance.
2. While the title of the manuscript indicates hydrogels based on “Schiff base” linkages, major content of the manuscript describes benzoic imine-based hydrogels, giving only limited information from the other kinds of hydrogels, e.g. the oxime and hydrazine linked ones.
3. Several important citations (and corresponding discussions) are missing. For example, while the authors focused on benzoic imine-based hydrogels, the very first work that reports such hydrogels did not mention (Ding et al. Biomacromolecules 2010, 11, 1043).
4. For the application, the authors just introduced what the hydrogels could do, but failed to discuss the advantage(s) of the Schiff base linked hydrogels in biomedical applications. Many gels can achieve injectability, self-healing property and can be also potentially used in drug delivery and tissue engineering, but what’s difference in comparison with the Schiff base linked hydrogels?
5. There are many typos in the text, including the author name line.
Author Response
It is very much appreciated that the reviewers made valuable comments. Our replies to the comments and the improvement of the manuscript based on the comments are listed below.
---------------------------------------------------------------------------------------
Reviewers’ Comments to Author:
1. The authors classified Schiff’s reaction under the catalog of Click chemistry. This might be wrong. Click chemistry represents the selective reactions that can rapidly join small modular units together in high yield and without offensive byproducts. In most of the cases, Schiff’s reaction is reversible, that is, the formation and hydrolysis are in dynamic balance.
Ans. Thank you for the professional suggestion. We have already supplemented and rewritten this part. The most correct and precise definition you mentioned has been added to the first sentence of the paragraph. However, there are many recent literatures categorizing Schiff base linkages into click chemistry (such as Journal of Materials Chemistry B, 5(23), 4435-4442; Nature Materials, 14(5), 523; Polymer Chemistry, 5(8), 2695-2699; Biomaterials, 35(18), 4969-4985; Biomacromolecules, 13(10), 3013-3017; Macromolecular rapid communications, 38(19), 1700357; Polymer chemistry, 5(11), 3555-3558; Biomacromolecules, 16(7), 2101-2108; Polymer Chemistry, 7(23), 3812-3826).
2. While the title of the manuscript indicates hydrogels based on “Schiff base” linkages, the major content of the manuscript describes benzoic imine-based hydrogels, giving only limited information from the other kinds of hydrogels, e.g. the oxime and hydrazine linked ones.
Ans. We have provided more examples of oxime- and hydrazine-based hydrogels. Moreover, we have modified some words to explain the Schiff base linkages and benzoic Schiff base linkages (imines). Schiff base linkages are mainly divided into three kinds, including imines, hydrazones, and oximes based on different types of amines. Benzoic Schiff base (imine) is a family of bonding formed by aldehyde groups connected with benzene rings and amine, hydrazine, or aminooxy groups, including benzoic imine, benzoic oxime, and benzoic hydrazine. Because benzoic Schiff base (imine) has become much more popular in recent studies, benzoic Schiff base (imine) is introduced in this manuscript using an individual paragraph.
3. Several important citations (and corresponding discussions) are missing. For example, while the authors focused on benzoic imine-based hydrogels, the very first work that reports such hydrogels did not mention (Ding et al. Biomacromolecules 2010, 11, 1043).
Ans. We have added some citations into the manuscript, particularly, the first and important work reported by Ding et al. in 2010. Not all earlier works were included because we tried to refer the papers published in the recent five years. We have also included many important papers by the group of Qu.
4. For the application, the authors just introduced what the hydrogels could do, but failed to discuss the advantage(s) of the Schiff base linked hydrogels in biomedical applications. Many gels can achieve injectability, self-healing property and can be also potentially used in drug delivery and tissue engineering, but what’s difference in comparison with the Schiff base linked hydrogels?
Ans. Thank you for the suggestion. We have added a paragraph (lines 233-242) in the biomedical application section to discuss the advantages of hydrogels with Schiff base linkages, compared to hydrogels formed by other click chemistry mechanisms.
5.There are many typos in the text, including the author name line.
Ans. Thank you for the notice. We have further proofread to avoid errors/typos. The errors you mentioned did not appear in the first edition of the manuscript that we submitted. We will communicate and confirm with the editorial office.
Round 2
Reviewer 2 Report
The quality of the manuscript was improved by the revision. Therefore, it is recommended for publication in the journal.